# Three-Dimensional Osteogenic Differentiation of Bone Marrow Mesenchymal Stem Cells Promotes Matrix Metallopeptidase 13 (MMP13) Expression in Type I Collagen Hydrogels

**DOI:** 10.3390/ijms222413594

**Published:** 2021-12-18

**Authors:** Luis Oliveros Anerillas, Paul J. Kingham, Mikko J. Lammi, Mikael Wiberg, Peyman Kelk

**Affiliations:** 1Department of Integrative Medical Biology, Umeå University, 901 87 Umeå, Sweden; luis.oliveros@umu.se (L.O.A.); paul.kingham@umu.se (P.J.K.); mikko.lammi@umu.se (M.J.L.); mikael.pj.wiberg@umu.se (M.W.); 2Department of Surgical & Perioperative Sciences, Section for Hand and Plastic Surgery, Umeå University, 901 87 Umeå, Sweden

**Keywords:** biogel, cell differentiation, mesenchymal stem cells, MMP13, MSCs, osteogenesis, type I collagen, 3D culture

## Abstract

Autologous bone transplantation is the principal method for reconstruction of large bone defects. This technique has limitations, such as donor site availability, amount of bone needed and morbidity. An alternative to this technique is tissue engineering with bone marrow-derived mesenchymal stem cells (BMSCs). In this study, our aim was to elucidate the benefits of culturing BMSCs in 3D compared with the traditional 2D culture. In an initial screening, we combined BMSCs with four different biogels: unmodified type I collagen (Col I), type I collagen methacrylate (ColMa), an alginate and cellulose-based bioink (CELLINK) and a gelatin-based bioink containing xanthan gum (GelXA-bone). Col I was the best for structural integrity and maintenance of cell morphology. Osteogenic, adipogenic, and chondrogenic differentiations of the BMSCs in 2D versus 3D type I collagen gels were investigated. While the traditional pellet culture for chondrogenesis was superior to our tested 3D culture, Col I hydrogels (i.e., 3D) favored adipogenic and osteogenic differentiation. Further focus of this study on osteogenesis were conducted by comparing 2D and 3D differentiated BMSCs with Osteoimage^®^ (stains hydroxyapatite), von Kossa (stains anionic portion of phosphates, carbonates, and other salts) and Alizarin Red (stains Ca^2+^ deposits). Multivariate gene analysis with various covariates showed low variability among donors, successful osteogenic differentiation, and the identification of one gene (matrix metallopeptidase 13, *MMP13*) significantly differentially expressed in 2D vs. 3D cultures. MMP13 protein expression was confirmed with immunohistochemistry. In conclusion, this study shows evidence for the suitability of type I collagen gels for 3D osteogenic differentiation of BMSCs, which might improve the production of tissue-engineered constructs for treatment of bone defects.

## 1. Introduction

Despite the innate capabilities of physiological regeneration of bone tissue, a current challenge in the fields of orthopedics and odontology is the reconstruction of large bone tissue defects due to trauma, and congenital and other pathological conditions, such as cancer [1,2].

The current gold standard for bone reconstruction is autologous bone grafting with or without attached vascularization. This approach, albeit effective, is associated with elevated donor-site infection, acute pain, and morbidity [3,4]. In addition, the mass of bone that can be grafted is limited. Thus, several artificial materials have been used as bone substitutes with variable results [5]. The field has further advanced by utilizing novel therapies with biomaterial scaffolding, in vitro creation of cellular constructs and use of decellularized bone for scaffolding [6,7].

Mesenchymal stem cells (MSCs), also called multipotent stromal cells, are postnatal cells that maintain physiological cell replenishment. MSCs can be isolated from a plethora of tissues, among which bone marrow MSCs (BMSCs) are the most common and extensively studied [8,9]. These cells can be utilized for treatment of various conditions and diseases, and globally there are currently 500 ongoing trials using BMSCs [https://clinicaltrials.gov, accessed date: 16/July/2021]. Their bioavailability and differentiation potential make them suitable for developing advanced cell therapies with or without scaffolding made of natural and biosynthetic matrixes for biocompatible transplantation [8,10,11,12,13].

For half a century MSCs have been isolated and cultured in vitro as adherent cells in order to expand and study the cells [14]. Since a three-dimensional (3D) environment offers multiple advantages compared with the standard 2D cell culturing methods, the elaboration of 3D constructs has been a trend in the field recently. Due to the higher complexity of 3D cultures, cells should display characteristics more akin to the natural cell–cell and cell–extracellular matrix (ECM) interactions found in vivo. Typically, MSCs are differentiated in vitro by the addition of various factors in order to push them to a specific cell lineage. However, in a 3D, more in vivo-like culture setup, BMSCs can be osteogenically differentiated by compressive forces in type I collagen in the presence of hydroxyapatite (HA) and β-tricalcium phosphate (βTCP) without the need for additional factors [15,16,17,18].

In order to make MSC-based cell therapy for bone regeneration a clinical reality and for this to become an alternative to autologous bone transplantation, there is a need to study and expand the cells in a more in vivo-like environment and combine the cells with natural or synthetic biocompatible materials. Several studies have shown the importance of 3D culturing of MSCs [19,20,21]. However, direct comparison between 2D and 3D culture conditions from the same MSC donors are lacking. In addition, various biomaterials and biogels that are utilized in combination with MSCs are not clinically approved [5].

In this study, our aim was to make a direct comparison of 2D and 3D in vitro cultures of BMSCs with matched donors with a focus towards mainly the osteogenic cell lineage. We evaluated various commercially available biogels that can be combined with BMSCs for osteogenic differentiation.

## 2. Results

### 2.1. Characterization of Donor Cells

The BMSC cultures were analyzed by flow cytometry for various stem cell surface markers. The majority of cells expressed CD73, CD90 and CD105 (>95%), while only subpopulations of the cells expressed CD146 (Figure 1A,C). The cells lacked expression of negative markers CD11b, CD19, CD34, CD45 and HLA-DR (Figure 1B,C).

The cumulative number of population doublings were assessed over a period of 128 days (Figure 1D). The BMSCs from the second youngest donor showed the highest rate of proliferation (donor 2; age 19), while the oldest (donor 1; age 65) showed the slowest proliferation rate. The other two donors (donor 3, age 38; and donor 4, age 17) showed intermediate proliferation rates. These results were consistent with the colony-forming unit assay (Figure 1E) in which single cells from donor 2 formed the highest number of colonies in contrast to donor 1.

The differentiation capacity of the cells towards the three typical lineages: osteogenic, adipogenic and chondrogenic, was determined (Figure 1F). 

### 2.2. Gel Selection and Cell Number Optimization

Next, we focused on establishing optimal conditions for osteogenic differentiation in 3D. Four commercially available biogels were tested for their ability to create a 3D construct in which the BMSCs could be differentiated. We selected two collagen-based biogels, one as pure type I collagen and the other methacrylated type I collagen (Col I and ColMa respectively). Further, we selected an alginate and nanocellulose-based gel (CELLINK) and a methacrylated gelatin and alginate-based gel supplemented with HA-TCP (GelXA Bone). The BMSCs were cultured in the biogels for 5 weeks in undifferentiated and osteogenic culture conditions. Only the type I collagen-based biogels both maintained their structural integrity and permitted the typical fibroblastic morphology of BMSCs (Figure 2A).

The rationale for using the ColMa was to reduce the expected long-term shrinkage of the constructs due to interaction between cells and collagen fibers. However, we did not observe any noticeable differences in shrinkage between Col I and ColMa gels (data not shown) and since the presence of methacrylate could potentially influence the cell functions, we decided to perform the rest of the study using only Col I gels.

There was a correlation between the number of cells and the shrinkage of collagen-based gels. We performed a short-term culture analysis to determine a suitable cell number that would not lead to unnecessary shrinkage but yield a sufficient amount of RNA for gene analyses. The preferred biogel, Col I, was cast in a 48-well plate to yield an initial 0.8 cm^2^ construct with various cell densities and cultured for 6 days to evaluate the shrinkage (Figure 2B,C). Although, the increased cell density resulted in shrinkage of the Col I gels, it reached a plateau at 1–5 × 10^5^ cells/gel. Subsequently, 3 × 10^5^ cells/gel were considered as a suitable cell number to avoid excessive shrinkage while maximizing sample yield for further analysis.

### 2.3. Osteogenic Differentiation in 3D

We next studied the osteogenesis of all four donor´s cells under the optimized 3D conditions. Histological staining with Osteoimage^®^ (HA-specific staining), von Kossa (stains anionic portion of phosphates, carbonates, and other salts) and Alizarin Red (stains Ca^2+^ deposits) showed that differentiation occurred in the Col I—BMSC cultures (Figure 3A). Furthermore, the 5 week differentiated cultures showed expression of osteogenesis-specific proteins osteocalcin (OCN, Figure 3B) and osteopontin (OPN, Figure 3C).

### 2.4. Time Course and Quantitative Analysis of Osteogenic Gene Expression

In order to further analyze the osteogenic differentiation from the four donors, we performed experiments at different time points in 2D or 3D. First, we determined that there was only a small variance between differentiation of the various donor cells. This was achieved by conducting a multivariate analysis with nSolver^®^ software using RNA samples from all groups versus the donor variable (volcano plots in Appendix A).

Then we proceeded to analyze the time course (after 1, 3 and 5 weeks) of gene expression differences due to osteogenic differentiation. Of the 66 tested genes we found that *SPP1, MMP13, COL10A1* were the most upregulated genes (Figure 4A), and *COL5A1, COL14A1, FGF2* were the most downregulated genes (Figure 4B). The results also showed that the most important time point to study osteogenic differentiation was at 5 weeks. Therefore, we performed a multivariate analysis in which gene differences were analyzed based on the differentiation at 5 weeks as the selected covariate (Figure 4C). The volcano plot shows a summary of significant gene changes (*p*-value < 0.01) highlighted in blue for downregulated genes (*COL15A1, COL14A1, COL12A1, FGF2*), and in green for upregulated genes (*COL10A, COL7A1, SPP1, MMP13*). 

### 2.5. Enhanced MMP13 Expression Due to 3D Osteogenic Differentiation

Finally, we wanted to investigate whether there were any differences between our 2D or 3D conditions from the matched donors. Hence, an additional multivariate analysis was conducted with the 2D or 3D conditions as covariates (Figure 5A). Interestingly, the only significant (*p*-value < 0.01) gene difference among the 66 tested genes was observed for *MMP13* (Figure 5A). The relative gene expression analysis of *MMP13* showed no significant difference between 2D and 3D undifferentiated groups (Figure 5B). However, there was significant upregulation of *MMP13* expression between 2D and 3D differentiated groups (*p*-value = 0.0008, Figure 5B). In addition, high statistical significance was demonstrated between the 3D undifferentiated and differentiated group (*p*-value = 0.0071). 

We determined that the gene expression changes correlated with increased protein expression. Immunostaining showed the clear presence of the MMP13 protein in differentiated 3D cultures (Figure 5C).

### 2.6. Other Cell Lineages

Since our results clearly indicated the enhanced potential for osteogenic differentiation in 3D, we also compared adipogenic and chondrogenic differentiation in 2D vs. 3D For adipogenesis, the 2D culture was an adherent cell culture and the 3D culture was made in Col I. For chondrogenesis, pelleted cultures (the standard technique for chondrogenesis in vitro) were used as the control model and the modified 3D culture was made in Col I. The results indicated that Col I 3D differentiation is a good candidate for adipogenesis (Figure 6A,B), while the traditional pellet method for chondrogenesis was superior compared with the Col I cultures (Figure 6C,D). 

## 3. Discussion

The potential and benefits of using MSCs in regenerative medicine for clinical applications has been established by a large number of studies. MSCs have low immunogenicity and are effective immune suppressors [22,23]. It has been identified than MSCs play a role in diverse organs to replenish and maintain tissue homeostasis and regenerate damaged tissue. The body has the potential to regenerate itself to some extent, but if the damage is too large, the regeneration is partially or completely hindered. Autologous bone cell transplantation is an established way to solve this problem, but this method has its limitations due to donor site availability, morbidity, and risk of infection. An alternative method to autologous bone transplantation is in vitro/ex vivo tissue engineering, which aims to regenerate tissues by combining cells with scaffolds and bioactive molecules. This field has advanced greatly in recent years. New advanced technologies of bioadditive manufacturing allow the construction of structures closer in architecture to bone tissue [24]. However, many of the biogels used to create 3D cultures contain substances that are not approved for clinical use. Despite this challenge, the advancement of tissue engineering must include 3D culturing in order to mimic physiological conditions of the target tissue that needs to be regenerated. In fact, there are a growing number of studies showing the superiority of 3D cultures compared with 2D cultures, for studying various disease models and also in the field of tissue engineering [25]. Despite displaying similar good results regarding cell morphology and gel appearance in our hands, ColMa gel was turned down in favor of unmodified Col I due to its known reduction of cell viability [26]. Consequently, in this study, we focused on establishing a reliable and reproducible 3D culture for osteogenic differentiation by testing different commercially available biogels, of which Col I gel was the optimal.

Type I collagen is the most abundant component of bone ECM [26] and so it has been regarded as the most obvious choice to create bone-regenerating constructs. However, working with collagen is not simple since its polymerization requires controlled pH and temperature conditions that make it difficult to 3D-print. However, a number of new techniques, such as the freeform reversible embedding of suspended hydrogels (FRESH) method, have been used to address this problem [27]. Furthermore, the natural shrinkage of collagen biogels remains a problem that has hindered the development of collagen-based constructs for bone regeneration [28,29]. In this study, we also report the shrinkage of collagen biogels as a factor to be reconsidered in tissue engineering. Despite these issues, collagen and mineralized constructs remain as one of the most promising candidates to treat bone defects in which the size of the gap impairs the natural regeneration of bone [30]. 

Bone marrow derived stem cells have clear advantages for transplantation and rebuilding bone tissue due to their endogenous role in healing damage. Two types of bone regeneration mechanisms exist. Primary regeneration occurs due to minor damage with no gap between proximal and distal ends of a fracture and involves the remodeling ECM without immune response. Secondary regeneration occurs when the ends of a fracture are separated and severe damage to the blood vessels has caused a hematoma and thus immune response. Blood leakage into the ECM constitutes a hematoma and triggers an inflammation response by infiltrating neutrophils. It has been reported that neutrophils recruit specific bone macrophages (osteomacs) to remove the temporary fibrin matrix and recruit BMSCs by means of specific inflammatory cytokines [31,32,33]. The presence of tissue damage promotes the migration of further MSCs to the injury site and the reduced inflammation leads to fibrous tissue production [11,34]. In addition, macrophages release osteogenic and proliferative cytokines. This process results in a disorganized ECM containing high numbers of both BMSCs and osteogenic monocytes that will take part in the remodeling and mineralization of the fibrous tissue along with the promotion of angiogenesis. Consequently, one of the main groups of secreted factors in this process are the matrix metalloproteases (MMPs) [35,36,37].

Intriguingly, in this study one of the MMPs, *MMP13*, was the only significantly affected gene out of the 66 tested osteogenic genes as a consequence of differentiation in 3D. MMPs are a family of zinc-dependent enzymes, with a total of 23 different proteins, whose primary function is ECM remodeling. Surprisingly, more than half of the family are expressed in their active form by bone and cartilage cells under both physiological and pathological conditions. MMPs can be classified according to their structural and substrate affinity. However, the most important MMPs for bone development are MMP2, MMP9, MMP13, MMP14 and MMP16 [36,38]. Due to the importance of MMPs for osteogenesis, recent publications have shown that MMP-13 upregulation is associated with osteogenic differentiation of human MSCs and even that the treatment with recombinant MMP-13 can also drive the differentiation process. Osteogenic differentiation can be initiated by activating the positive feedback loop involving runt-related transcription factor 2 (RUNX2), integrin α3 (ITGA3) and focal adhesion kinase and MMP13 promoter [37,39]. In addition, MMP13 is essential to perform peri-lacunar remodeling and maintain the resistance against bone fracture [40]. MMP13 is also upregulated in hypertrophic chondrocytes as a downstream target of Cbfa1/Runx2, which results in its implication in bone mineralization and bone resorption [41]. 

Some research groups have also experimented with the in vitro addition of recombinant MMPs and BMPs to enhance matrix remodeling and mineralization capabilities of BMSCs [42,43,44]. Similarly, the addition of recombinant platelet derived growth factor, basic fibroblast growth factor (bFGF), and vascular endothelial growth factor to increase the angiogenic potential, has been considered a promising method to induce osteogenic differentiation [45].

In summary, this study shows evidence for suitability of type I collagen gels for 3D osteogenic differentiation of BMSCs. The 3D osteogenic differentiation of BMSCs promotes MMP13 expression in type I collagen-based hydrogels. Our further studies aim to combine this 3D culturing method with novel osteo-inductive materials. These findings also strengthen the ability of 3D constructs to be utilized clinically in the future.

## 4. Materials and Methods

### 4.1. Tissue Harvest, Cell Isolation and Culture

The local ethics committee for research at Umeå University (Dnr 2013-276-31M and 03-425) approved collection, processing, culture, storage, and usage of all clinical isolates in this study. All methods were performed by following the relevant guidelines and regulations of the local ethics committee. Informed consent was obtained from all donors. 

Healthy bone marrow tissue was collected from the iliac crest or ulna from four human donors (mean age 34.75 years; 17–65 years). Samples were rinsed thoroughly with minimum essential medium-α (α-MEM; Invitrogen) containing 10% (*v*/*v*) fetal bovine serum (FBS; Sigma Aldrich, St. Louis, MO, USA) and 1% (*v*/*v*) penicillin–streptomycin (Gibco). The cell suspension was centrifuged at 300× *g* for 5 min, the cell pellet was then filtered through a 70 μm nylon mesh (BD Falcon) and cells were then plated in the above growth medium in 75 cm^2^ tissue culture flasks (Nunc) and incubated at 37 °C with 5% CO_2_. After 24 h in culture, the supernatant containing non-adherent cells was removed and fresh medium added. The cells attached to the culture flask were cultured for 2–3 weeks with medium changes every 48 h. When the cultures had reached 80–90% confluence, the cells were enzymatically detached with 0.05% trypsin/EDTA (Gibco) by incubating for 5 min at 37 °C, followed by centrifugation at 200× *g* for 5 min, washed with Dulbecco’s phosphate buffered saline (D-PBS) and frozen in 90% FBS/10% DMSO solution. Samples were stored in liquid nitrogen until required. Cells were then thawed at 37 °C and seeded at a cell density of 1 × 10^4^ cells/25 cm^2^ flask, initially in α-MEM supplemented with 20% FBS and 10 ng/mL of bFGF (PeproTech, London, UK). Once the cell population reached 90% confluence, they were detached with trypsin/EDTA (Gibco) and passaged to new 75 cm^2^ flasks at a cell density of 5000 cells/cm^2^. 

### 4.2. Cell Proliferation and Colony-Forming UNIT Assay

Cells from passage 1 were seeded at 10^4^ cells per well in a 6-well plate, in triplicates, per donor. The growth medium used was α-MEM supplemented with 10% FBS and 1% penicillin/streptomycin, which was changed every other day. Once one of the cultures (from any donor) had reached 80–90% confluence, all the individual cell populations were enzymatically detached, counted with a hemocytometer and re-seeded in a new 6-well plate with 10^4^ cells/well. The cells were expanded for 128 days in total. Population doublings (PDs) were calculated using the following equation:PD=[duration ×log(2)]/[log(Final cell number)−log(Initial cell number)]

In addition, BMSCs from all four donors at passage 1 were plated for the CFUf assay; 300 cells were seeded into a 25-cm^2^ flask and cultured for 2 weeks without medium change, followed by fixation, and staining with a 0.1% (*w*/*v*) toluidine blue, 2% (*w*/*v*) paraformaldehyde solution for 1 h. The flasks were rinsed with distilled water to remove the dye excess. Only colonies with ≥50 cells were counted as a CFUf.

### 4.3. Cell Characterization by Flow Cytometry

BMSCs at passage 2 were examined for the expression of MSCs-associated CD markers (CD73, CD90 and CD105) and a negative marker cocktail (CDCD11b, CD19, CD34, CD45 and HLA-DR). According to the manufacturer´s protocol (BD Biosciences, Allschwil, Switzerland), BMSCs were incubated with phycoerythrin (PE)-conjugated antibodies (CD73 (1:25), CD90 (1:33) or CD105 (1:25)) and negative marker cocktail (CD11b, CD19, CD34, CD45 and HLA-DR (1:25)). PE mouse IgG1κ isotype control (BD Pharmingen) was used as control for all positive CD markers. PE human MSC (hMSC) isotype control, negative cocktail (BD Stemflow) was used as an isotype negative control. The antibody list is displayed at Table 1. A total of 5 × 10^4^ cells for each donor were collected per antibody and 10^4^ were analyzed after SSC/FFC-gating using an Accuri™C6 Plus flow cytometer (BD Bioscience) as previously described [46].

### 4.4. Commercial Gel Selection

A set of four different commercially available biogels were initially selected. These were Col I (an unmodified type I collagen), ColMa (type I collagen methacrylate, a UV-crosslinkable gel), CELLINK (an alginate and cellulose-based bioink), and GelXA-bone (a gelatin-based bioink containing xanthan gum). Col I and ColMa gels consisted of a final collagen concentration of ~4–4.4 mg/mL. All the gels were acquired from CELLINK and combined with 5 × 10^5^ BMSCs per gel. A set of gels without cells were also created and used as negative control. All gels were cast following the manufacturer´s instructions in a 48-well plate. ColMA gels were, in addition to the regular crosslinking, also exposed to UV radiation (405 nm) for 45 s to enhance polymerization. CELLINK and GelXA Bone gels were produced following the manufacturer´s protocol and polymerized with addition of CaCl_2_.

### 4.5. Gel-Cell Optimization

BMSCs from passage 3 were used to evaluate the shrinkage due to cell population ratio in the gels. On ice, acidic collagen stock solution at 5 mg/mL (CELLINK) was mixed 10:1 with a neutralizing solution (5X PBS, 0.1N NaOH), vortexed and it´s neutral pH verified before the addition of the cells. Final collagen concentration was of ~4–4.4 mg/mL and the casting performed in 48-well culture plates. The volume of each gel was set at 200 μL and then incubated for 30 min at 37 °C to assure proper collagen gelation. The cell number per gel ranged from 10^4^ to 5 × 10^5^. Once the collagen was solid, the gels were washed with D-PBS and then incubated in 300–400 μL Dulbecco’s modified eagle medium (DMEM; Gibco) culture media supplemented with 10% FBS and 1% penicillin-streptomycin solution. The gels were incubated for 6 days before they were scanned on a transparent plastic sheet, in order to evaluate how shrinkage is dependent on the cell number. The optimized cell number was set to 3 × 10^5^ cells/gel to maximize RNA and protein quantity while minimizing gel shrinkage.

### 4.6. Gel Casting, Differentiation, and Harvest

Prior to the casting, the BMSCs from all four donors were trypsinized, counted, and mixed with a neutralized solution of collagen (~4–4.4 mg/mL, CELLINK) on ice, to prevent undesired polymerization, followed by their casting onto a 48-well plate. Then, the plate was incubated for 30 min at 37 °C to assure proper collagen gelation. The number of cells in each gel was set to 3 × 10^5^. Cell culture medium was added and after 1 day of incubation, the gels were moved from a 48-well plate to a 24-well plate to maximize medium exposure. Next, 2D control cultures were plated at 3 × 10^5^ cells/well to match the cell number in 3D groups. The 2D groups were plated as adherent cells as previously described [47] for osteogenic and adipogenic experiments. The 2D cultures used for chondrogenic differentiation were pelleted into 5 mL tubes to allow pellet formation. 

The differentiation process started 24 h after the gels were cast and 2D cultures were plated by exposing the cultures to differentiation media: 

Osteogenic differentiation, DMEM (low glucose + Glutamax), 10% FBS, 1% (*v*/*v*) penicillin–streptomycin, 100 nM dexamethasone, 10 mM β-glycerophosphate and 0.2 mM 2-phosphate-L-ascorbic acid.

Adipogenic differentiation, DMEM (low glucose + Glutamax), 10% FBS, 1% (*v*/*v*) penicillin–streptomycin, 1 µM dexamethasone, 10 µg/mL insulin, 0.5 mM 3-isobutyl-1-methylxanthine, and 100 μM indomethacin.

Chondrogenic differentiation, serum-free α-MEM supplemented with 40 µg/mL L-proline, 100 µg/mL sodium pyruvate, 100 µM dexamethasone, 1% ITS +3 liquid media supplement, 50 µg/mL ascorbate 2-phosphate and 10 ng/mL TGF-β3.

The differentiation and control media were changed every other day and samples collected after 1, 3 and 5 weeks in the case of osteogenesis, 4 weeks for adipogenesis and 5 weeks for chondrogenesis. The 3D and 2D-culture samples for each differentiation condition were processed at equal time points. For every experimental group, RNA was extracted for both 2D and 3D conditions for real time-PCR analysis to measure the expression levels of typical differentiation markers of the three cell lineages. In order to further confirm the differentiation, samples were either stained with a lineage-specific dye or fixed in paraformaldehyde, cryosectioned (12-μm-sections) and used for immunohistochemistry and visualized with a bright field and fluorescence microscope.

### 4.7. Histochemistry

Histochemical staining to confirm osteogenic, adipogenic and chondrogenic differentiation was performed as described below. The 2D culture staining was performed in culture well-plates, while 3D cultures were initially cryosectioned and mounted on glass slides as described below.

Osteogenesis was analyzed with specific staining. Osteoimage^®^ (Lonza, Walkerville Inc., Australia) staining was performed following the manufacturer´s instruction adapted for small size specimens. Von Kossa staining was performed by following the kit manufacturer´s protocol (ab150687, Abcam, Cambridge, UK). Pre-filtered 1% Alizarin red (Thermo Fisher Scientific, Gothenburg, Sweden)) solution was added for 3 min to previously washed sections (with PBS) and fixed with 60% isopropanol and rehydrated with deionized water. The dye excess was washed with distilled water before mounting the sections with PERTEX^®^ on glass slides for imaging.

Adipogenesis was tested with Oil red O dye. Sections were washed with PBS and fixed with 60% isopropanol followed by incubation with Oil red O for 10 min at room temperature. Thereafter, a final wash was performed with 60% isopropanol and samples rehydrated with PBS.

Chondrogenic differentiation was assessed by Masson´s trichrome staining (HT15-1KT, Sigma-Aldrich) by following manufacturer´s protocol and using Weigert´s Iron Haematoxylin (HT1079, Sigma) to counter-stain the cells. 

Haematoxylin-Eosin (HE) staining was performed by using Meyer hematoxylin solution and following manufacturer´s instructions.

### 4.8. Immunohistochemistry (IHC)

The slides were washed twice with IHC buffer (0.01 M PBS, 0.1% NaN_3_ and 0.1% BSA) for 30 min and blocked with 5% goat normal serum (diluted in IHC buffer). Excess buffer was removed, and the slides incubated overnight at 4 °C with primary antibody (Table 2) dissolved in IHC buffer, followed by 4 washes of 15 min with IHC buffer and a secondary block with 5% goat normal serum in IHC buffer. The slides were incubated in the dark for 1 h with the secondary antibody (Table 2), diluted into IHC buffer, and washed with buffer. Sections were mounted with DAPI Prolong mounting medium (P36935, Life Technologies).

### 4.9. Semi-Quantitative and Quantitative RT-PCR

Total RNA was extracted from undifferentiated and differentiated 2D and 3D cultures of BMSCs by using a RNeasy mini kit according to the manufacturer’s instructions (Qiagen). The RNA was quantified with a Nanodrop-2000c spectrophotometer (ThermoFisher Scientific). In the case of semi-quantitative reverse transcription-PCR (RT-PCR), 1 ng of RNA were incorporated into a One-Step RT-PCR kit (Qiagen) per reaction mix and PCR products were determined with 3% (*w*/*v*) agarose gel electrophoresis separation. For quantitative (q) RT-PCR, 250 ng complementary DNA (cDNA) were synthesized using an iScript cDNA synthesis kit (Bio-Rad, Hercules, CA, USA). Per reaction, 5 ng of cDNA were used, along with SsoFast™EvaGreen^®^ supermix (Bio-Rad, Solna, Sweden) in a CFX96 Optical Cycler and analyzed using the CFX96 manager software (Bio-Rad). Ribosomal protein L13a (RPL13a) was used as housekeeping gene and the data calculated as relative expression according to the ∆∆C(t) principle. The following primers were used (Table 3).

### 4.10. NanoString nCounter Human Bone-Related Gene Expression Analysis

NanoString™ nCounter^®^ is a molecular barcode-based hybridization reaction analysis that is used as an alternative to both qRT-PCR and mRNAseq to determine level of gene expression in small sample sizes. The NanoString™ system was used to analyze 2D and 3D culture BMSCs with or without osteogenic stimulation mRNA samples by using their nCounter^®^ analysis system. In this study, a custom-made kit was designed against 66 human endogenous osteogenesis-related genes. The majority of the selected genes were bone-related, with addition of some embryonic stem cell- and angiogenic-related genes. A summary of the genes are is as follows: *ALPL, BGLAP, BMP1, BMP2, BMP3, BMP4, BMP5, BMP6, BMP7, BMPR1A, BMPR1B, CALCR, COL1A1, COL1A2, COL2A1, COL3A1, COL4A3, COL4A4, COL4A5, COL5A1, COL7A1, COL9A2, COL10A1, COL11A1, COL12A1, COL14A1, COL15A1, COL16A1, COL17A1, COL18A1, COL19A1, DSPP, FGF1, FGF2, FGF3, FGFR1, FGFR2, FGFR3, FLT1, GDF10, ISBP, MMP13, MMP2, MMP8, NANOG, PDGFA, PHEX, RUNX2, SMAD1, SMAD2, SMAD3, SMAD4, SMAD6, SMAD6, SMAD7, SMAD9, SOX9, SPP1, TGFB1, TGFB2, TGFB3, TGFBR1, TGFBR2, VEGFA, VEGFB,* and *VEGFC*.

These genes were normalized in the nCounter system against 6 housekeeping genes (for accession and target sequences see Appendix A).

The first set of samples consisted of twelve pooled BMSCs from four donors cultured in 2D or 3D for 1, 3 and 5 weeks in control or osteogenic condition. The second set of samples were sixteen individual RNA solutions (originating for four different donors; n = 4; N = 16) of BMSCs cultured in 2D or 3D for five weeks in control or osteogenic condition.

After both sets were run by nCounter^®^ analysis system, the data were analyzed using nSolver^®^ software and presented as boxplots or volcano plots as a gene expression ratio.

### 4.11. Statistical Analysis

In this study, we performed multi-variate analysis by one-way ANOVA followed by post hoc Bonferroni test (GraphPad Prism^®^, GraphPad Software Inc., San Diego, CA, USA). Bonferroni correction was also used in the nSolver software for the analysis of significant changes in the nanoString system. Statistical significance was set as ** *p* < 0.01, *** *p* < 0.001.

## Figures and Tables

**Figure 1 ijms-22-13594-f001:**
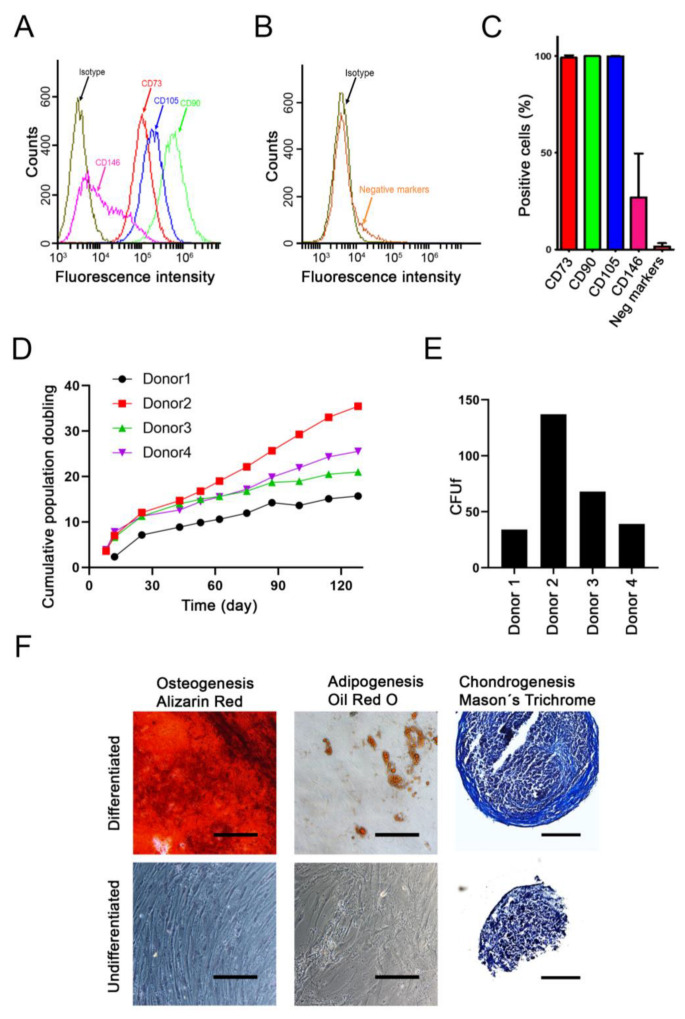
Characterization of donor MSCs. (**A**–**C**) Flow cytometry characterization of positive- and negative-associated CD markers. (**A**) A representative donor overlay plot is shown for the expression of positive markers: CD73, CD90, CD105 and CD146, in relation to the isotype control. (**B**) A representative donor overlay plot is shown for the expression of negative markers CD11b, CD19, CD34, CD45 and HLA-DR, in relation to the isotype control. (**C**) Quantification of the expression of positive and negative CD markers (mean + SD, *n* = 4). (**D**) Cumulative population doubling assessed for each of the donor cells for a total of 128 days. (**E**) Colony forming unit fibroblastic (CFUf) was determined after two weeks of culture with the starting density of 300 cells in a T25 flask. (**F**) Undifferentiated and differentiated cells from a representative donor are shown. Different staining was used to confirm the differentiation of the cells: for osteogenesis (5 weeks, Alizarin red), for adipogenesis (4 weeks, Oil red O), and for chondrogenesis (5 weeks, Masson’s trichrome). Scale bar = 200 μm.

**Figure 2 ijms-22-13594-f002:**
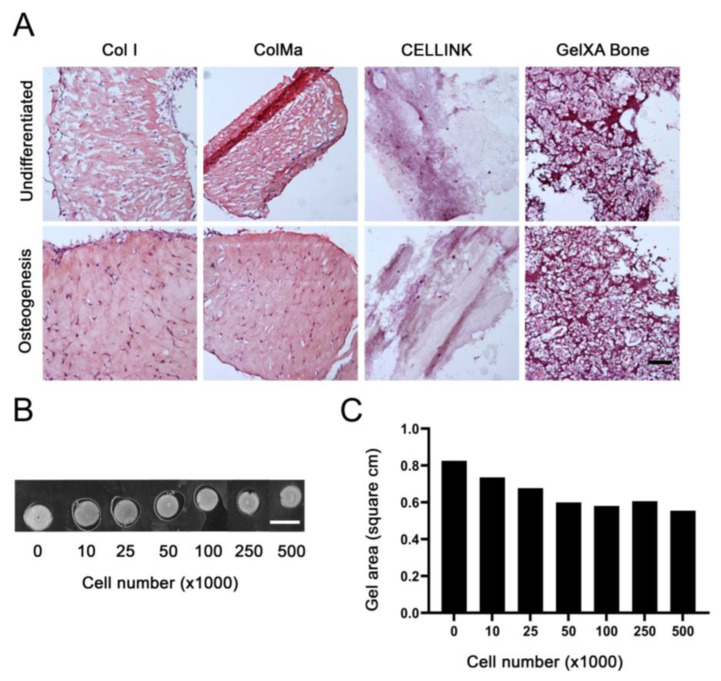
Gel selection and cell number optimization. (**A**) HE stainings from one BMSC donor cultured in 3D with different biogels: type I collagen (Col I), methacrylated type I collagen (ColMa), alginate and nanocellulose-based bioink (CELLINK) and a methacrylated gelatin and alginate-based gel supplemented with HA-TCP (GelXA Bone). The gels were cultured with 2 × 10^5^ BMSCs/gel for 5 weeks in control or osteogenic media. Scale bar = 100 μm. (**B**) Six days in vitro Col I gel cast at 200 μL to yield 0.8 cm^2^ constructs with various cell numbers per gel to visualize gel shrinkage. Scale bar = 1 cm (**C**) Quantification of scanned gel size displayed in B processed with Image J Software.

**Figure 3 ijms-22-13594-f003:**
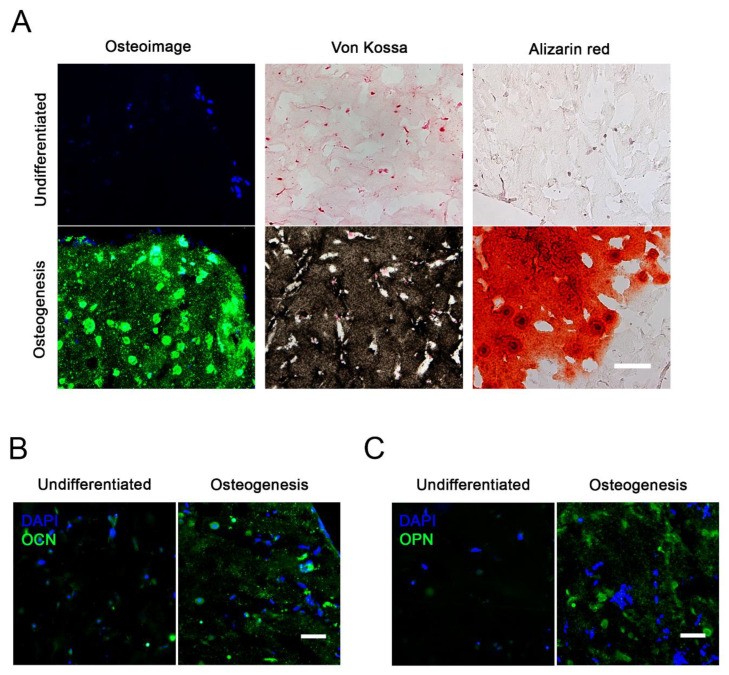
Osteogenic differentiation in 3D. (**A**) Histochemical staining of 12-μm-thick gel sections from one representative donor after 5 weeks of culture in undifferentiated or osteogenic conditions. Osteoimage^®^ (HA-specific staining), von Kossa (stains anionic portion of phosphates, carbonates, and other salts) and Alizarin Red (stains Ca^2+^ deposits) shows differentiation in the Col I—BMSC cultures. Scale bar = 100 μm. (**B**) Immunostaining for osteocalcin (OCN) and nuclear 4′,6-diamidino-2-phenylindole (DAPI) in 3D cultures after 5 weeks. (**C**) Immunostaining for osteopontin (OPN) and nuclear DAPI in 3D cultures after 5 weeks. Scale bar = 50 μm.

**Figure 4 ijms-22-13594-f004:**
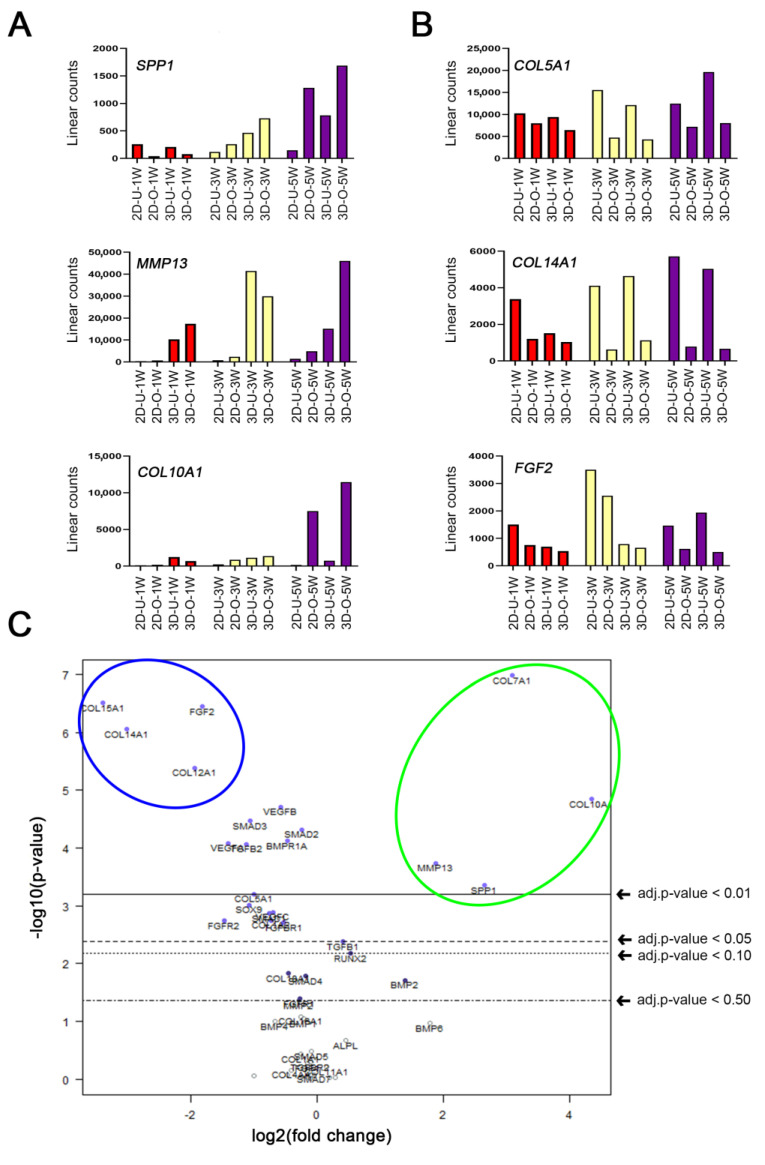
Time course and quantitative analysis of osteogenic gene expression. (**A**,**B**) Time course of pooled RNA samples from four different donors were screened by a NanoString^®^ kit for 66 (mainly osteogenic) genes after 1, 3 or 5 weeks of control or osteogenic condition. Week timepoints are shown as W1 (red), W3 (yellow), W5 (purple). Meanings: 2D = adherent cell cultures, 3D = Col I gel cultures, U = Undifferentiated cells in control medium, and O = differentiated cells in osteogenic medium. (**A**) Time course of the three most upregulated out of sixty-six tested genes: *SPP1, MMP13, COL10A1.* (**B**) Time course of the three most downregulated out of sixty-six tested genes: *COL5A1, COL14A1, FGF2*. (**C**) Multivariate analysis of the sixty-six tested genes, after 5 weeks of differentiation, displayed as a volcano plot (created with nSolver^®^ software) with the differentiation set as covariate. The plot shows each gene’s −log_10_ (*p*-value) and log_2_ fold change with the differentiation set as covariate. Highly statistically significant genes fall at the top of the plot above the indicated *p*-value lines, and highly differentially expressed genes fall to either side (right = upregulated, left = downregulated). The most significantly (*p*-value < 0.01) upregulated genes are highlighted in the green circle (*COL10A, COL7A1, SPP1, MMP13*), and the most significantly (*p*-value < 0.01) downregulated genes are highlighted in the blue circle *(COL15A1, COL14A1, COL12A1, FGF2*).

**Figure 5 ijms-22-13594-f005:**
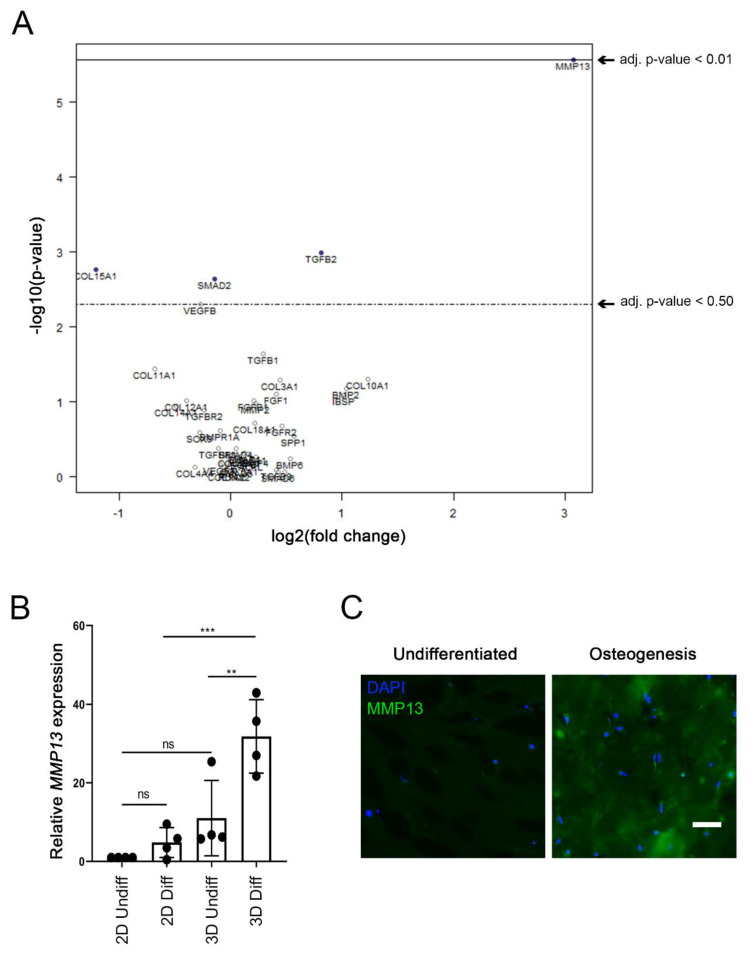
Enhanced MMP13 gene expression and production due to 3D osteogenic differentiation. (**A**) Multivariate analysis of the sixty-six tested genes (NanoString^®^ kit) displayed as a volcano plot (created with nSolver^®^ software) with the 2D or 3D condition set as covariate. The plot shows each gene’s −log_10_ (*p*-value) and log_2_ fold change associated to the 2D or 3D condition. Highly statistically significant genes fall at the top of the plot above the indicated *p*-value lines, and highly differentially expressed genes fall to either side (right = upregulated, left = downregulated). The only significantly (*p*-value < 0.01) affected gene that was upregulated was MMP13. (**B**) Relative gene expression of MMP13 is shown in the four various conditions: 2D undifferentiated, 2D differentiated, 3D undifferentiated, 3D differentiated, after 5 weeks of culture. The significance levels for relative gene expression difference are indicated as ** (*p*-value < 0.01), *** (*p*-value < 0.001). (**C**) Immunohistochemistry of 5-week undifferentiated vs. differentiated (osteogenic) 3D Col I gels against MMP13 and nuclear DAPI. Scale bar = 50 μm.

**Figure 6 ijms-22-13594-f006:**
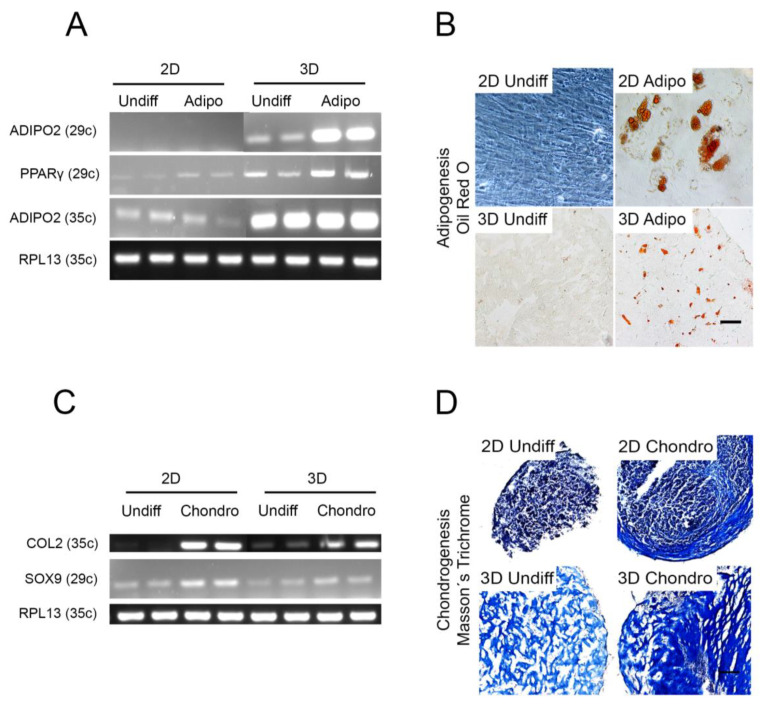
Adipogenic and chondrogenic differentiation in 2D and 3D. (**A**,**B**) BMSCs were cultured in four different conditions: 2D undifferentiated (2D undiff), 2D differentiated (2D diff), 3D undifferentiated (3D undiff), 3D differentiated (3D diff), in control or adipogenic medium for four weeks in 2D (adherent monocultures) or 3D (Col I gels, containing 3 × 10^5^ cells per gel). (**A**) Agarose gel electrophoresis of semi-qt RT-PCR analysis of relevant adipogenic genes: ADIPO2 and PPARγ at 29 and 35 amplification cycles (29c and 35c respectively). (**B**) Oil red O staining in 2D and 3D after adipogenic differentiation. (**C**,**D**) BMSCs were cultured in four different conditions: 2D undifferentiated (2D undiff), 2D differentiated (2D diff), 3D undifferentiated (3D undiff), 3D differentiated (3D diff), in control or chondrogenic medium for four weeks in 2D (pellet cultures) or 3D (Col I gels, containing 3 × 10^5^ cells per gel). (**C**) Agarose gel electrophoresis of semi-qt RT-PCR analysis of relevant chondrogenic genes: COL2 and SOX9 at 29 and 35 amplification cycles (29c and 35c, respectively). (**D**) Masson´s trichrome staining in 2D and 3D after chondrogenic differentiation. Section thickness of 3D models = 12 μm. Scale bar = 100 μm.

**Table 1 ijms-22-13594-t001:** Antibodies used for flow cytometry.

Antibody	Brand	Clone Number
CD73	BD Pharmingen™	Mouse IgG1, κ AD2
CD90	BD Pharmingen™	Mouse BALB/c IgG1, κ 5E10
CD105	BD Pharmingen™	Mouse BALB/c IgG1, κ, 266
CD146	BD Pharmingen™	Mouse IgG1, κ, P1H12
PE mouse IgG1, κ isotype control	BD Pharmingen™	Mouse IgG1, κ MOPC-21
PE hMSC negative cocktail	BD Stemflow™	Mouse IgG1, κ CD34 (Clone 581) CD11b PE (Clone: ICRE44) CD19 PE (Clone: HIB19) CD45 PE (Clone: HI30) HLA-DR PE (Clone G46–6)
PE hMSC negative isotype control cocktail	BD Stemflow™	Mouse IgG1, κ, PE (Clone × 40)

**Table 2 ijms-22-13594-t002:** Antibodies used for immunohistochemistry.

Type	Antibody	Reference	Dilution
Primary	Mouse monoclonal anti MMP13	R&D Biosystems, MAB511	1:50
Primary	Mouse monoclonal, anti-OCN	Novus Biologicals, H00000632-M01	1:100
Primary	Mouse monoclonal, anti-OPN	Novus Biologicals, NB110-89062	1:100
Secondary	Alexa FluorTM 488 Goat anti mouse	ThermoFisher, A11029	1:1000

**Table 3 ijms-22-13594-t003:** Primers used for semi- and quantitative RT-PCR analysis.

Primer	Forward (5′-3′)	Reverse (5′-3′)	Annealing Temperature (°C)
OPN	GCCGACCAAGGAAAACTCACT	GGCACAGGTGATGCCTAGGA	64.7
OCN	AGCAAAGGTGCAGCCTTTGT	GCGCCTGGGTCTCTTCACT	63.2
ADIPO2	GGTGGTGGAATGCGTCATG	CAACGTCCCTTGGCTTATGC	64.1
ALP	GGAACTCCTGACCCTTGACC	TCCTGTTCAGCTCGTACTGC	65.2
PPARγ	CCGAGAAGGAGAAGCTGTTG	TCGGATATGAGAACCCCATC	60.8
RPL13a	AAGTACCAGGCAGT GACAG	CCTGTTTCCGTAGCCTCATG	58
SOX9	GACTTCCGCGACGTGGAC	GTTGGGCGGCAGGTACTG	57.5
COL2A	AGACTGGCGAGACTTGCGTCTA	ATCTGGACGTTGGCAGTGTTG	57.3

## Data Availability

Not applicable.

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
