# Peer review of "Three-Dimensional Osteogenic Differentiation of Bone Marrow Mesenchymal Stem Cells Promotes Matrix Metallopeptidase 13 (MMP13) Expression in Type I Collagen Hydrogels"

_ijms, 2021, doi:10.3390/ijms222413594_

Round 1

Reviewer 1 Report

The current study is a very important research work in the field of bone biology, will attract mass attention. The manuscript is well-written. However, I have following suggestions:

  1. Line 81-82, donor 2 age 28……. The other two donors (age 28 and 47) showed intermediate proliferation rates. If age 28 was the youngest with highest proliferation and 74 was the oldest with slowest proliferation, and age 28 and 47 showed the intermediate proliferation, then the youngest (age 28) showed higher proliferation does not make any sense as both were age 28 with different rates. Please make corrections here.
  2. Line 89, data not shown. It seems to be interesting and important data. So please include them.
  3. Figure 1D, please use different colors for different donor lines for better illustration.
  4. Figure 1E, please show SD or SEM.
  5. Figure 1F, please include negative control (that stained negative in osteogenesis, adipogenesis, chondrogenesis) images for all staining.
  6. Figure 2C and other graphs where applicable, please show SD or SEM.
  7. Line 104 and in all the cases, please use Italic fonts whenever required, for example, in vitro.
  8. Line 300, 313, cells were enzymatically detached. Please mention the name of the enzyme and concentration, time etc. if applicable.
  9. Line 354, neutralized collagen solution. How was the solution neutralized?
  10. Figure S2 shows important data. Why was it placed in supplementary?
  11. Please add a specific conclusion and if possible, include a graphical abstract.

Reviewer 2 Report

This is an interesting and well-done study about 3D osteogenic differentiation of bone marrow mesenchymal stem cells promoting MMP 13 expression in type I collagen hydrogels. I suggest it for publication in IJMS after the following points are addressed.

  1. Line 121-122, why the presence of methacrylate influences the cell functions should be added? The author may add the ref. describing the effect of methacrylate group on gels.
  2. Line 129-130, it is not clear why the number of cells and the shrinkage of collagen-based gels has a direct relation. The author should add more discussion about this phenomenon.
  3. The resolution of figure 1a and 1b should be improved to a higher level.
  4. The legend in figure 4c and 5a are overlapped.
  5. Line 49-51, several recent studies (10.1021/acs.biomac.0c01641; doi.org/10.1016/j.actbio.2019.11.015) should be included to support such claim.
